

# Bioefficacy of engineered *Beauveria bassiana* with scorpion neurotoxin, LqqIT1 against cotton mealybug, *Phenacoccus solenopsis* and cowpea aphid, *Aphis craccivora*

Sonam Rajput[1,2], Sachin S. Suroshe[1], Purnmasi Ram Yadav[2], Anoop Kumar[3] and Gurvinder Kaur Saini[4]

[1] Biological Control Laboratory, Division of Entomology, ICAR-IARI, New Delhi, Delhi, India
[2] Zoology Department, DAV College (CCS University, Meerut), Muzaffarnagar, India
[3] ICAR-NCIPM, Pusa Campus, New Delhi, India
[4] Department of Biosciences and Bioengineering, IIT-Guwahati, Guwahati, India

Corresponding author
Sachin S. Suroshe,
sachinsuroshe@gmail.com

## ABSTRACT

Cotton mealybug, *Phenacoccus solenopsis* (Tinsley) and cowpea aphid *Aphis craccivora* (Koch) are notorious polyphagous, hemipteran sap sucking insect pests. A recombinant toxin gene 'LqqIT1' from the scorpion *Leiurus quinquestriatus quinquestriatus* (Ehrenberg) was cloned in the pAL1 fungal expression vector and then expressed in the entomopathogenic fungus *Beauveria bassiana* (Balasmo) using genetic modification techniques. The genetically transformed *B. bassiana* strain (BbLqqIT1-3) and its un-transformed parent strain (Bb-C) were screened to infect the third instar nymphs of *P. solenopsis* and first instar nymph of *A. craccivora* through leaf treatment and topical application (spray) method at $1 * 10^7$ spores per ml concentration. The recombinant strain BbLqqIT1-3 was highly pathogenic against *A. craccivora* but non pathogenic to *P. solenopsis*. BbLqqIT1-3 induced 72 and 43.33% mortality in *A. craccivora* nymphs 96 h after leaf treatment and topical application, respectively. The nymphs of *A. craccivora* infected with BbLqqIT1-3 displayed classical neurotoxic symptoms such as sluggishness, solublize and liquification of the body. Crude soluble toxin protein, BbLqqIT1a-CSE and Bb-WT-CSE was extracted from the BbLqqIT1-3 and Bb-C, respectively using ammonium sulphate precipitation method, and their oral toxicity was analyzed at 5 μg/ml concentration. The survival of the studied insects was negatively affected by the crude soluble toxin extracts. The $LT_{50}$ values of BbLqqIT1a-CSE against *P. solenopsis* and *A. craccivora* were 22.18 and 17.69 h, respectively. Exposure to crude soluble toxin extracts also accounted for the imbalance of ionic concentrations in the hemolymph of treated insects such as hyperpotassemia (3.53–8.18 meq/ml) in the *P. solenopsis* and hypopotassemia (7.52–0.47 meq/ml) in *A. craccivora*. The transformed fungus BbLqqIT1-3 strain exhibited promising results in invitro study.

## INTRODUCTION

*Phenacoccus solenopsis* (Tinsley) and *Aphis craccivora* (Koch) are the most economically important polyphagous insect pests prevalent in the Indian subcontinent. *P. solenopsis* is a well-known sap-sucking insect pest which attacks agricultural and vegetable crops all over the world (*Khuhro et al., 2012*). The phloem-feeding of host plants by this insect pest has been documented to inflict significant economic losses. *P. solenopsis* are generally found near to the terminal stem, leaves, and fruits of plants, however due to their cryptic lifestyle, which entails hiding in crevices, beneath the bark, leaf whorls, and on the roots where pesticides cannot reach them, they are challenging to control chemically (*Watson, 2016*). *P. solenopsis* have waxy substance around their body that repels water and inhibits water-based pesticide treatments from penetrating their bodies (*Hemalatha et al., 2018*). *A. craccivora* causes direct damage by feeding on plant phloem, as well as indirect damage through the excretion of honeydew and the transmission of viruses that cause leaf shrinkage and impaired growth (*Obopile, 2006*). Both immature and adult stages are accounted for the serious damage during all the stages of the plants (*Saranya et al., 2010*). Control of sucking insect pests is challenging due to their polyphagous nature, short life cycle and high reproductive rate (*Suresh, Khan & Prasanna, 2012*). Rapid killing speed and ability to kill any insect pests drives us to use broad spectrum synthetic chemicals as a first line of defense against dreaded insect pests such as *P. solenopsis* and *A. craccivora*, but this strategy is very limited due to development of insecticide resistance in aphids and mealybugs and their ill effects on the environment (*Jackai & Adalla, 1997*). The use of conventional pesticides to combat *P. solenopsis* and *A. craccivora* has been dominated much of the research. In terms of long-term chemical control of insect pests, entomopathogens are generally regarded as useful and promising. Entomopathogens such as entomopathogenic fungi (EPF) are one of the most well-known and promising biological control agents currently being studied and used. EPFs *viz.*, *Metarhizium anisopliae* (Metchnikoff) Sorokin, *Beauveria bassiana* (Balasmo) Vuillumin, *Lecanicillium lecanii* (Zimmermann), *etc.*, have been used extensively due to their pathogenicity and broad host range (*Thomas & Read, 2007*). The EPFs are little bit slow compared to synthetic insecticides in killing insect pests (*St Leger et al., 2009*). Therefore, the killing efficiency of EPFs could be improved for higher effectiveness through a variety of means, like increasing fungal toxin production and by incorporating insect-specific neurotoxins derived from scorpion/spider, which operate within the insects (*Peng & Xia, 2014*). A variety of EPFs have been shown to release insecticidal, antifeedant, and insect specific bioactive metabolites in liquid cultures which could be purified or extracted (*Quesada-Moraga, Carrasco-Díaz & Santiago-Álvarez, 2006*). The use of microbial bio-active metabolites in modulating insect behaviour has already been described by many researches (*Strasser, Vey & Butt, 2000*). Several researchers have reported the insecticidal activity of secondary metabolites *viz.*, beauvericins, bassianolides, bassiacridin, destruxins, oosporein, oxalic acid, and tenellin from EPFs (*Quesada-Moraga & Vey, 2004*; *Farooq & Freed, 2018*).

According to *Murugan & Saini (2019)*, rLqqIT1 is a sodium activated channel toxin which leads to rapid paralysis and physiological impairment in *Spodoptera litura*

(Fabricius) and *Helicoverpa armigera* (Hubner) larvae. By causing recurrent firing in the motor neurons, LqqIT1 causes acute spastic paralysis in the target species (*Eitan et al., 1990*). In this study, LqqIT1 gene was introduced and integrated into the *B. bassiana* to check whether the expression of LqqIT1 gene enhances the pathogenicity of *B. bassiana* towards third instar of *P. solenopsis* and first instar nymph of *A. craccivora*.

Hemolymph circulates in the insect body cavity and hence contributes considerably to proper physiological activities (*Gillott, 1995*). Many cellular activities are influenced by the direct or the indirect action of inorganic ions in bodily fluids (*Treherne, Buchan & Bennett, 1975*). Extensive research into the mode of action of secondary metabolites from the entomopathogenic bacteria such as *Bacillus thuringiensis* (Berliner) on insects revealed that it alters the ionic concentrations of hemolymph and the pH of the gut (*Tiwari & Mehrotra, 1981*). Keeping all these in view, the studies were planned to screen the enhanced pathogenicity of engineered *B. bassiana* (BbLqqIT1-3) against third instar nymph of *P. solenopsis* and first instar nymph of *A. craccivora* using two different bioassay methods *viz.*, leaf treatment and spray method, and to elucidate the toxicological effects of fungal crude soluble toxin extracts (BbLqqIT1a-CSE and Bb-WT-CSE) on *P. solenopsis* and *A. craccivora* and, to evolve the change in the hemolymphic inorganic cationic ($K^+$ and $Na^+$) activity of the *P. solenopsis* and first instar nymph of *A. craccivora* when treated with the BbLqqIT1a-CSE and Bb-WT-CSE.

## MATERIALS AND METHODS

The present experiments were conducted in the Biological control laboratory of Division of Entomology, ICAR- IARI, New Delhi ICAR-IARI, Pusa, New Delhi. Model insect populations were maintained at $27 \pm 1$ °C and $65 \pm 5\%$ RH with a 16 h of photoperiod in environmental control chamber throughout the course of experimentation.

### Insect collection, rearing and maintenance

Initially, *P. solenopsis* was collected from the China rose, *Hibiscus rosa- sinensis* (Linnaeus) and cotton field, *Gossypium hirsutum* (Linnaeus) from the fields of the division of Entomology, ICAR-IARI, Pusa campus, New Delhi. Then, reared on sprouted potato tubers and subsequent generations were used for the experiment. The freshly moulted third instar nymphs were used for the bioassay (*Suroshe, Gautam & Fand, 2016*).
*A. craccivora* were collected from the infested cowpea (*Vigna unguiculata* (Linnaeus) Walpers) plants from the fields of ICAR-IARI, Pusa, New Delhi. Aphids were reared on to the 3–5 days old, fresh and tender cowpea seedlings grown in the plastic jars (15.5 cm Θ) using absorbant cotton (*Kumar et al., 2023*). This population was maintained as a stock culture throughout the study period. First instar nymphal stage (≥24 h old) was used for bioassay.

### Fungus culture

Genetically engineered *B. bassiana* along with parent (untransformed) strain was procured from Dr. Gurvinder Kaur Saini, Professor, Department of Biosciences and Bioengineering, IIT Guwahati, Assam, India and multiplied in the Biological Control Laboratory, ICAR-

IARI, Pusa, New Delhi for conducting pathogenic experiments on selected insect pests. Originally, *B. bassiana* (MTCC 984, untransformed (Bb-C)) culture was obtained from the Microbial Type Culture Collection and Gene Bank (MTCC, Chandigarh, India), and it was kept on SDA (Sabroud Dextrose Agar) medium at 28 °C. After incubating for 7–10 days on SDA, spores were collected.

## Generation of the recombinant strain (BbLqqIT1-3)
Following techniques were followed for the development of recombinant *B. bassiana*

### LqqIT1 gene synthesis
Protein sequence of LqqIT1 was obtained from National Center for Biotechnology Information (NCBI) database with the accession number: P19856.1 followed by its reverse translation into DNA sequence using European Molecular Biological Laboratory (EMBL)-EMBOSS Backtranseq software (*Rice, Longden & Bleasby, 2000*). The reverse translated DNA sequence LqqIT1 was codon optimized for bacterial expression. Further, the codon optimized DNA sequence was synthesized and cloned in plasmid-pUC57 flanking Eco RI and *Hin*dIII restriction sites (Genscript, Piscataway, NJ, USA) (*Murugan & Saini, 2019*).

### LqqIT1 codon optimization for expression in *B. bassiana*
Based on the native toxic protein sequence, corresponding gene sequence was codon optimized for the expression in *B. bassiana*. In addition to the toxic gene sequence 5′ untranslated region (UTR, Kozak sequences) and Mcl1 signal peptide (Mcl1ss) was included at the 5′ region from *Metarhizium* collagen like protein, to transport the protein of interest to hemolymph. The sequence was codon optimized using *Beauveria bassiana* codon usage table (http://www.kazusa.or.jp/codon/). Codon optimized gene was synthesized (Genescript, Piscataway, NJ, USA) and cloned in pUC57 cloning vector (2.7 kbp) within KpnI and Bam HI restriction sites. The size of the codon optimized gene (376 bp) was confirmed by transforming the vector pUC57 harboring LqqIT1 gene into *Escherichia coli* (Castellani & Chalmers) DH5α chemical competent cells, and then isolated plasmids were subjected to double digestion with KpnI and Bam HI restriction enzymes.

### Vector construction and fungal transformation
Along with LqqIT1, 5′ untranslational region and *Metarhizium* collagen like protein signal peptide sequence was also codon optimized and cloned in pUC57 vector which resulted in pUC-5′UTR-Mcl1sp-LqqIT1 plasmid which was further confirmed by restriction enzyme digestion and PCR amplification of LqqIT1 gene sequence. After codon optimization and cloning, 5′ UTR-Mcl1sp-LqqIT1 fragment was amplified, and fused downstream of *Metarhizium* collagen like protein promoter (PMcl1) to express the LqqIT1 protein in soluble form in *B. bassiana* when induced with hemolymph conditions. To deliver the protein into the insect cavity during the infection process, LqqIT1 was tagged N-terminally with *Metarhizium* collagen like protein signal peptide sequence (Mcl1-sp) along with 5′ untranslational region (5′ UTR). Expression was confirmed by restriction enzyme

digestion and PCR confirmation. In this way pMcl1-LqqIT1 expression system was developed.

The pAL1-PMcl1-LqqIT1a (3–10 μg) plasmid was transformed into *B. bassiana* using protoplast cum electroporation mediated transformation method and fungal protoplast was prepared following the protocol by *Pfeifer & Khachatourians (1987)* with modifications. Vector pMcl1-LqqIT1 (3–10 μg) was mixed with 150 μl of *B. bassiana* protoplast ($1 * 10^8$ protoplast/ml) and incubated on ice 15–30 min in 0.2 cm electroporation cuvette. A nine hundred voltage was used in transferring the plasmid DNA into protoplast. Colony growth was observed after 10 days post selection plating in Czapek Dox agar ($K_2HPO_4$-1 gm/l, $FeSO_4.7H_2O$-0.01 gm/l, $MgSO_4.7H_2O$-0.5 gm/l, KCl- 0.5 gm/l, $NaNO_3$-3 gm/l, sucrose- 30 gm/l, $(NH_4)_2SO_4$ and 200 μg/ml) glufosinate-ammonium (Sigma-Aldrich, St. Louis, MO, USA).

## Analysis of genomic integration

To verify the expression of the Mcl1ss-LqqIT1 fused gene within the hemolymph of insect, total RNA was extracted from mycelia on the SDB medium using optimized extraction procedure (RNAiso plusKit; Takara, Shiga, Japan). One microgram of total RNA was used for cDNA synthesis (Superscript[TM] III first strand synthesis system, Invitrogen, Carlsbad, CA, USA). Synthesized cDNA was used for semi quantitative RT-PCR analysis based on LqqIT1 specific primers. Integration of LqqIT1 transgene into the genomic DNA of *B. bassiana* was tested by analysing the inducible expression of transgene in hemolymph using semi quantitative RT-PCR and western blot.

## Sub-culturing of engineered *B. bassiana*

Genetically transformed *B. bassiana* was subcultured as per the previously described method (*Kumar et al., 2023*). Procured fungal source (aqueous fungal suspension) was subcultured on standard Sabouraud Dextrose Agar medium (SDA) (Himedia) in full darkness at 27 ± 1 °C and 80 ± 5% RH. Conidial suspension for the pathogenicity test was obtained from 25 to 30 days old fungal colonies. Conidia were harvested using 10 ml sterile distilled water containing 0.02% Tween 80 and filtered through sterile cheese cloth to remove mycelial mat (*Chan-Cupul et al., 2010*). Obtained filtrates were preserved at 4 °C for further bioassay and extraction of crude soluble toxin extract (CSE). Before using the preserved aqueous fungal suspension was shaken vigorously. The conidial concentration of the resulting 'stock' suspension was estimated using an improved Neubauer hemocytometer (Marienfeld) under a Nikon ECLIPSE 80i microscope (400 × magnifications) and number of spores present per ml was estimated using the formula by *Aneja (1996)*.

## Extraction of crude filtrate

### Production and extraction of crude soluble toxin extract (CSE) in Czapek Dox broth medium

Crude soluble toxic extract (CSE) was extracted as per the method described by *Quesada-Moraga, Carrasco-Díaz & Santiago-Álvarez (2006)* and *Farooq & Freed (2018)* with some
modifications. Czapek Dox liquid (enriched with 1% yeast and 0.5% peptone) medium was used to make a primary culture. All processes for the extraction of crude toxin were carried out at 4 °C.

### Protein concentration of crude soluble toxic extract

Protein concentration of the toxic crude extracts was estimated by the Bradford method using Bovine serum albumin (BSA) as standard (*Bradford, 1976*). Absorbance was measured at 280 nm (*Quesada-Moraga & Vey, 2004*).

### Bioassay procedure

The entomopathogenic activity of un-transformed (Bb-C) and transformed (BbLqqIT1-3) strain and their crude soluble extract (BbLqqIT1a-CSE and Bb-WT-CSE) was evaluated and compared using two methods. All fungal experimental trials were carried out under aseptic conditions inside a laminar air flow chamber sterilized with UV radiation. An improved Neubaur Hemocytometer under a compound microscope at 400 × magnification was used to arrive at number of spores present per ml as per the formula described by *Aneja (1996)*.

## Leaf treatment method (for fungus and crude soluble extract)
### Cotton mealybug, P. solenopsis

The virulence of the transformed fungal strain was examined by leaf treatment method (*Insecticide Resistance Action Committee, 2009*) (Method No-019) with a little modification. Fresh cotton leaves with petioles were cut from the potted plants (25–30 days old), then washed thoroughly with tap water and then allowed to dry in the shade. Thousand microlitre of testing solutions (fungal suspensions at $1 * 10^7$ spores/ml and crude soluble extract at 5 µg/ml) was applied onto both sides of the leaf (1,000 µl testing solution per leaf) after shade drying on paper towel, and then kept on the agar bed (1.5%) in Petri dish (90 mm Θ). On each, 10 third-instar nymphs were released. Mortality was measured every 24 h interval up to 168 h after treatment.

### Cowpea aphid, A. craccivora

The pathogenicity of the transformed fungal strain was examined by leaf treatment dip bioassay method with the suitable modification (*Moores et al., 1996*). We selected cowpea leaves that were in the trifoliate stage. Before treatment, leaves were rinsed with sterile water, dipped in the $1 * 10^7$ spores/ml fungal aqueous suspension (2.5 ml for five leaves) and then allowed to dry on sterile filter paper at room temperature. After that, the treated leaves were placed upside down on an agar bed (1.5%) in Petri dish (90 mm Θ). Ten first instar nymphs were released in each Petri dish using a soft hairbrush of zero size. Mortality was noted every 24 h up to 168 h after treatment.

## Spray method (only for fungus)
### Cotton mealybug, P. solenopsis

Cotton leaves (25 to 30 days old) were used for the treatment. To increase the life span of the leaves, the base of the petiole was wrapped with sterile, damp cotton. Fresh cotton

leaves were placed in Petri dish (90 mm Ө) and 2 h starved third nymphs of *P. solenopsis* were released using a soft hairbrush @ 10 per leaf. Once the nymphs got settled, each was sprayed with fungal spore suspension (1 $^*$ 10$^7$ spores/ml) (10 ml for five leaves) using an atomizer. Control batch of nymphs were sprayed with sterile distil water having 0.02% Tween 80 (positive control) and untransformed parent strain (Bb-C). Fungus treated insects were incubated in separate growth chamber @ 28 ± 1 °C and 80 ± 5% RH. At each 24 h interval up to 7 days (168 h), mortality, mobility of treated insects and fungal development was examined and recorded. At each 24 h interval or as per the need old leaves were replaced by fresh and untreated leaves.

### Cowpea aphid, A. craccivora

A total of 5 day old cowpea seedlings were used for the experiment. One day old nymphs were released on the cowpea seedlings using zero size camel hairbrush @ 10 per seedling. Once the aphids settled, the fungal spore suspension (1 $^*$ 10$^7$ spores/ml and 10 ml for five seedlings) was sprayed using an atomizer. Old and wilted seedlings were replaced with fresh and untreated one. Mortality was measured at each 24 h interval up to 168 h after spray. Dead nymphs were immediately placed in the sterile Petri plate with damp sterile filter paper for mycosis and sporulation. Each experiment was carried out independently. The experiment was repeated four times. Five independent replications with thirty insects per replication were maintained for each treatment. The treated insect was declared dead if no coordinated movement was noticed to the external stimulus (*i.e.*, when gently touched with a fine paint brush) under Nikon stereo binocular microscope having camera attached with the software VIMAGE 2016.

## Sample preparation for flame photometry (cationic activity)

To reveal the impact of crude soluble toxin on hemolymphic Na$^+$ and K$^+$ levels, we assessed changes at 24, 48, 72 and 96 h after treatment. We examined hemolymph samples collected from treated insects. For that, 100 µg of treated insects were taken in 2 ml eppendroff tube and homogenized by using hand held plastic pestle in 900 µl of phosphate buffer saline. The homogenate was centrifuged at 10,000 rpm for 15 min at 4 °C. The supernatant was collected and used as hemolymph source for the estimation of cations. Change in ionic concentration was estimated by the method of *Malik & Malik (2009)* with a little modification. One ml of supernatant (hemolymph), 10 ml of diacid in 9:4 ratio (nitric acid and percholic acid) was added and was digested using digestion chamber until a clear solution (volume 2–5 ml) was obtained. The solution was filtered through Whatman No-1 filter paper. The cations were measured from the aliquots of filtrate. Sodium and potassium ions were measured by using flame photometer (SYSTRONIC 128$^{µc}$). The change in ionic concentration was analyzed at 48, 72 and 96 h after treatment at 5 µg/ml concentration of crude extract. Cationic concentration was expressed as millimole per microlitre (mm/ml) or (meq/ml) per ml of hemolymph, using sodium chloride (Nacl) and potassium chloride (Kcl) as standards.

## Statistical analysis

The experiment was conducted in a Completely Randomized Design (CRD). The data was analysed by probit analysis (*Finney, 1971*) and the median lethal concentration ($LC_{50}$) and the median lethal time ($LT_{50}$) values were estimated by using SPSS version 21 (*IBM Corp, 2012*). The mean values of hemolymphic cations were subjected to WASP (version 2.0) online available statistical tool and the mean differences between the treatments were tested by ANOVA at 5% level of significance and the level of significance ($\alpha$) was set at 0.05 ($P \leq 0.05$).

The percent reduction in population was calculated by using Henderson and Tilton's formula (*Henderson & Tilton, 1955*).

## RESULTS

Biotoxicity of un-transformed (Bb-C) and transformed (BbLqqIT1-3) strain and their crude soluble extract (BbLqqIT1a-CSE and Bb-WT-CSE) was screened against the nymphs of *P. solenopsis* and *A. craccivora* in term of the $LC_{50}$ and $LT_{50}$. The toxic effect increased with the increment in the exposure time as presented in below sections. However, some natural mortality was observed in control batch also which was corrected by using Abbott's formula (*Abbott, 1925*). It was noted that presence of toxic gene LqqIT1 provoked a hyperpotassemia in the *P. solenopsis* while in *A. craccivora* hypopotassemia was seen.

### Efficacy of transformed *B. bassiana*, BbLqqIT1-3 against *P. solenopsis*

The recombinant strain BbLqqIT1-3 integrated with scorpion neurotoxin LqqIT1 was evaluated against third instar *P. solenopsis* nymph *via* leaf treatment and spray method. BbLqqIT1-3 caused 6.67 and 36.67% mortality after 72 and 168 h of treatment whereas it was 13.33 and 46.67%, respectively in untransformed strain (Bb-C) (Fig. 1). *P. solenopsis* treated with BbLqqIT1-3 by leaf treatment showed $LT_{50}$ of 188.13 h, which was 1.09-fold higher than that of mealybug treated with the un-transformed strain (Bb-C) (171.9 h) ($\chi^2 = 2.9$, df = 2, $P > 0.05$) (Table 1). In the leaf treatment and spray experiments, *P. solenopsis* treated with BbLqqIT1-3 had lower mortality than *P. solenopsis* treated with the untransformed strain (Bb-C) (Tables 1 and 2, Fig. 2) at 168 h after treatment. Same trend was observed in spray method.

### Efficacy of transformed *B. bassiana* 'BbLqqIT1-3' against *A. craccivora*

The results demonstrated that the survival of first instar nymph of *A. craccivora* infected with BbLqqIT1-3 was significantly lower than that of those infected with un-transformed strain 'Bb-C' at 7 day after treatment (98.11 ± 3; 26.84 ± 7.94) at constant spore load $1 * 10^7$ spores/ml (Table 3). Highest per cent reduction in the survival of *A. craccivora* nymphs was imposed by BbLqqIT1-3. The mean mortality rate increased with the increase in the time interval. In leaf treatment method, BbLqqIT1-3 strain produced 50% mortality at 96 h of treatment; while Bb-C required more than 168 h to achieve the same results (Fig. 3). Likewise, in spray method BbLqqIT1-3 strain caused 53.37% mortality after 96 h of treatment, which was 1.60-fold higher as compared to the Bb-C (33.34) (Fig. 4).

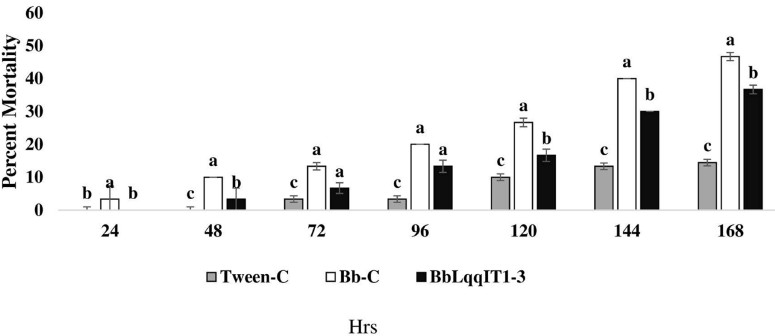

**Figure 1 Mortality of *P. solenopsis* third instar nymphs post infection of fungal strains Bb-C and BbLqqIT1-3 of *B. bassiana* via leaf treatment.** The error bars represents the standard error of the mean, different lowercase letters at the top of each bar shows significant differences using Duncan's multiple range test (DMRT) at 5% level.

**Table 1 Mean cumulative mortality and $LT_{50}$ after infection of genetically transformed and un-transformed strains of *B. bassiana* in *P. solenopsis* third nymphs *via* leaf treatment.**

| Strain @ $1 \times 10^7$ spores/ml | Mean cumulative mortality (Mean ± SE) | $LT_{50}$ (h) | Slope ± SE | Fiducial 95% (LL–UL) | $\chi^2$ value | P-value |
|---|---|---|---|---|---|---|
| Tween-C | 14.45 ± 0.6 | 519.25 | 2.36 ± 0.7 | 467.5–644.17 | 4.509 | 0.105 |
| Un-transformed Bb-C | 46.67 ± 0.7 | 171.9 | 0.011 ± 0.001 | 157.8–192.2 | 1.33 | 0.52 |
| Transformed (BbLqqIT1-3) | 36.66 ± 0.77 | 188.13 | 0.013 ± 0.002 | 173.09–210.5 | 2.9 | 0.24 |

Note:
Values are presented in Mean ± SE (standard error of the mean), every % mortality value represents the mean of the five independent replications, df = 2, lower $LT_{50}$ equivalent to higher pathogenecity; $LT_{50}$, median lethal time (hours); fiducial limit at 95%; LL, lower limit; UL, upper limit; $\chi^2$, chi square value: $P \leq 0.05$ indicate significant differences among the treatment.

**Table 2 Mean cumulative mortality and $LT_{50}$ after infection of genetically transformed and un-transformed strains of *B. bassiana* in *P. solenopsis* third nymphs *via* spray method.**

| Strain @ $1 \times 10^7$ spores/ml | Mean cumulative mortality Mean ± SE | $LT_{50}$ (h) | Slope ± SE | Fiducial 95% (LL–UL) | $\chi^2$ value | P-value |
|---|---|---|---|---|---|---|
| Tween-C | 15.96 ± 0.504 | 517.9 | 2.73 ± 1.44 | 418.106–717.44 | 1.2 | 0.549 |
| Un-transformed Bb-C | 26.66 ± 0.503 | 219.8 | 0.010 ± 0.001 | 193.7–265.3 | 5.69 | 0.065 |
| Transformed (BbLqqIT1-3) | 16.67 ± 0.49 | 247.43 | 0.010 ± 0.002 | 212.82–315.4 | 5.09 | 0.79 |

Note:
Values are presented in Mean ± SE (standard error of the mean), every % mortality value represents the mean of the five independent replications, df = 2, lower $LT_{50}$ equivalent to higher pathogenecity, $LT_{50}$, median lethal time (hours); fiducial limit at 95%; LL, lower limit; UL, upper limit; $\chi^2$, chi square value; $P \leq 0.05$ indicate significant differences among the treatment.

The $LT_{50}$ value in leaf treatment method was 2.56 h for BbLqqIT1-3 strain which was 5.58 times shorter compared to parent 'Bb-C' (14.26 h) ($\chi^2 = 8.7$, df = 2, $P < 0.05$) (Table 3). Similarly, the $LT_{50}$ after spray was 3.51 h for BbLqqIT1-3 strain which was about 1.76 times less compared with Bb-C (6.15 h) ($\chi^2 = 9.24$, df = 2, $P < 0.05$) (Table 4). Perusal of the

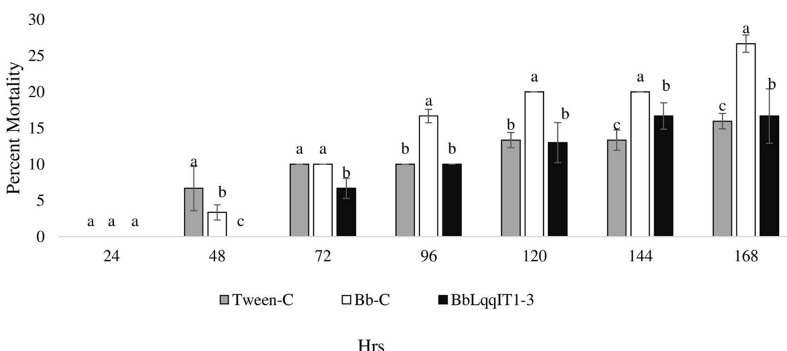

**Figure 2 Mortality of *P. solenopsis* third instar nymphs post infection of fungal strains Bb-C and BbLqqIT1-3 of *B. bassiana* via spray method.** The error bars represents the standard error of the mean, different lowercase letters at the top of each bar shows significant differences using Duncan's multiple range test (DMRT) at 5% level.

**Table 3 Mean cumulative mortality and $LT_{50}$ after infection of genetically transformed and un-transformed strains of *B. bassiana* in *A. craccivora* nymphs via leaf treatment.**

| Strain @ $1 \times 10^7$ spores/ml | Mean cumulative mortality Mean ± SE | $LT_{50}$ (h) | Slope ± SE | Fiducial 95% (LL–UL) | $\chi^2$ value | P-value |
|---|---|---|---|---|---|---|
| Tween-C | 17.79 ± 2.23 | 238.62 | 2.196 ± 0.607 | 162.09–765.9 | 4.92 | 0.086 |
| Un-transformed Bb-C | 26.84 ± 7.94 | 14.26 | 1.73 ± 0.443 | 6.91–25.29 | 5.029 | 0.0809 |
| Transformed (BbLqqIT1-3) | 98.11 ± 3.00 | 2.56 | 2.13 ± 0.307 | 1.50–5.81 | 8.7 | 0.013 |

**Note:**

Values are presented in Mean ± SE (standard error of the mean), every % mortality value represents the mean of the five independent replications, df = 2, lower $LT_{50}$ equivalent to higher pathogenicity, $LT_{50}$, median lethal time (hours); fiducial limit at 95%; LL, lower limit; UL, upper limit; $\chi^2$, chi square value; $P \leq 0.05$ indicate significant differences among the treatment.

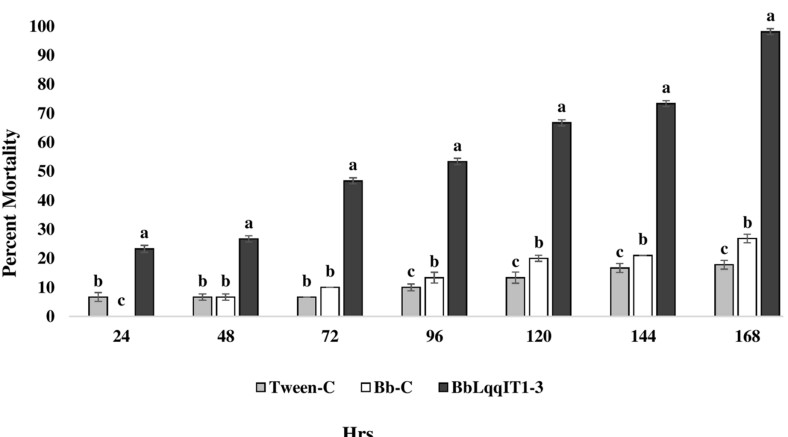

**Figure 3 Mortality of *A. craccivora* first instar nymphs post infection of fungal strains Bb-C and BbLqqIT1-3 of *B. bassiana* via leaf treatment.** The error bars represents the standard error of the mean, different lowercase letters at the top of each bar shows significant differences using Duncan's multiple range test (DMRT) at 5% level.

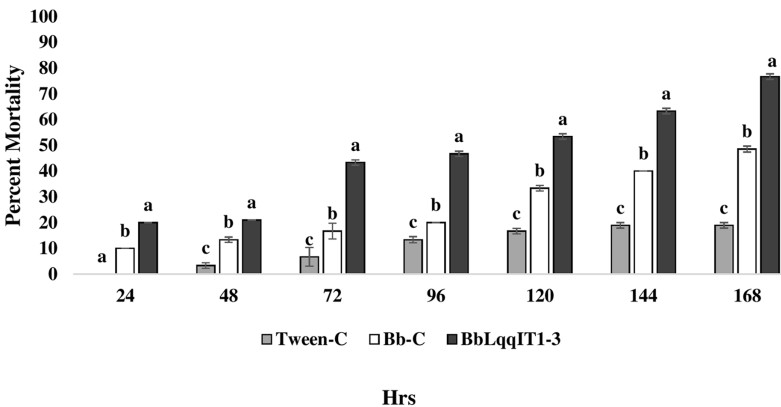

**Figure 4 Mortality of *A. craccivora* first instar nymphs post infection of fungal strains Bb-C and BbLqqIT1-3 of *B. bassiana* via spray method.** The error bars represents the standard error of the mean, different lowercase letters at the top of each bar shows significant differences using Duncan's multiple range test (DMRT) at 5% level.

**Table 4 Mean cumulative mortality and LT$_{50}$ after infection of genetically transformed and un-transformed strains of *B. bassiana* in *A. craccivora* nymphs via spray method.**

| Strain @ $1 \times 10^7$ spores/ml | Mean cumulative mortality Mean ± SE | LT$_{50}$ (h) | Slope ± SE | Fiducial 95% (LL–UL) | $\chi^2$ value | P-value |
|---|---|---|---|---|---|---|
| Tween-C | 18.89 ± 2.94 | 249.9 | 2.36 ± 0.698 | 167.5–986.8 | 0.698 | 0.705 |
| Un-transformed Bb-C | 48.56 ± 2.5 | 6.15 | 1.85 ± 0.32 | 3.94–15.82 | 5.96 | 0.05 |
| Transformed (BbLqqIT1-3) | 76.66 ± 2.2 | 3.51 | 1.60 ± 0.304 | 2.85–4.97 | 9.24 | 0.009 |

**Note:**
Values are presented in Mean ± SE (standard error of the mean), every % mortality value represents the mean of the five independent replications, df = 2, lower LT$_{50}$ equivalent to higher pathogenecity, LT$_{50}$, median lethal time (hours); fiducial limit at 95%; LL, lower limit; UL, upper limit; $\chi^2$, chi square value: $P \leq 0.05$ indicate significant differences among the treatment.

pathogenicity data revealed that the improved mortality by rLqqIT arised after the fungus entered in the host. These results confirmed that the integration of recombinant toxin gene (LqqIT1) improved the lethality of *B. bassiana* towards the nymphs of *A. craccivora*.

## Efficacy of BbLqqIT1a-CSE and Bb-WT-CSE against the *P. solenopsis* and *A. craccivora*

The LT$_{50}$ in *P. solenopsis* exposed with BbLqqIT1a-CSE was 22.18 h ($\chi^2$ = 9.418, df = 2, $P < 0.05$), which was 1.5-fold less compared to Bb-WT-CSE (32.14 h) in *P. solenopsis* ($\chi^2$ = 8.7, df = 2, $P < 0.05$) (Table 5). Treatment with BbLqqIT1a-CSE produced 59.74% mortality at 72 h of treatment in *P. solenopsis*; while Bb-WT-CSE produced only 36.77% mortality, which was 1.62-fold higher as compare to the Bb-WT-CSE (Fig. 5). At the end of experiment (seventh day), the maximum per cent mortality (83.34) was recorded in *A. craccivora* by BbLqqIT1a-CSE, whereas only 46.7% mortality was observed by Bb-WT-CSE (Fig. 6). The LT$_{50}$ values for *A. craccivora* treated with BbLqqIT1a-CSE and Bb-WT-CSE were 17.69 ($\chi^2$ =7.93, df = 2, $P < 0.05$) and 51.75, respectively ($\chi^2$ = 5.94, df = 2,

**Table 5 LT$_{50}$ of crude soluble toxic extracts of *B. bassiana* against *P. solenopsis* third instar nymphs after 72 h *via* oral bioassay.**

| Treatments @ 5 µg/ml of protein | LT$_{50}$ (h) | Slope ± SE | Fiducial (95%) (LL–UL) | $\chi^2$ value | *P*-value |
|---|---|---|---|---|---|
| PBS-C | 79.7 | 3.88 ± 0.7 | 34.59 ± 105.5 | 5.609 | 0.06 |
| Bb-WT-CSE | 32.14 | 3.68 ± 0.99 | 24.74–67.34 | 8.7 | 0.013 |
| BbLqqIT1a-CSE | 22.84 | 4.707 ± 1.37 | 18.39–46.25 | 9.418 | 0.009 |

Note:
Values are presented in Mean ± SE (standard error of the mean), df = 2, lower LT$_{50}$ equivalent to higher pathogenecity, LT$_{50}$, median lethal time (hours); fiducial limit at 95%; LL, lower limit; UL, upper limit; $\chi^2$, chi square value; $P \leq 0.05$ indicate significant differences among the treatment.

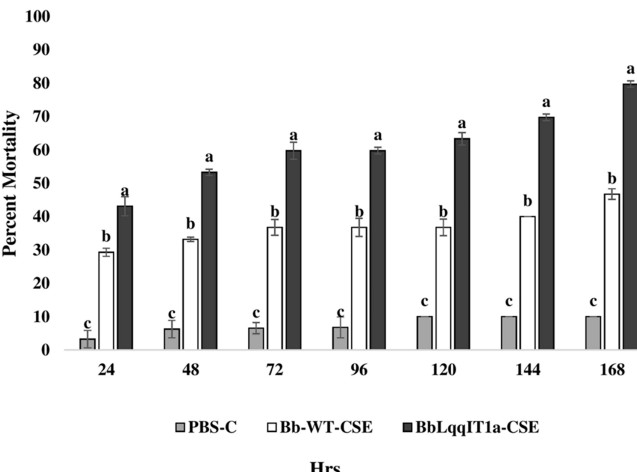

**Figure 5 Mortality of third instar nymphs of *P. solenopsis* after treatment of crude soluble toxin extract (CSE) @ 5 µg/ml.** The error bars represents the standard error of the mean, different lower-case letters at the top of each bar shows significant differences using Duncan's multiple range test (DMRT) at 5% level

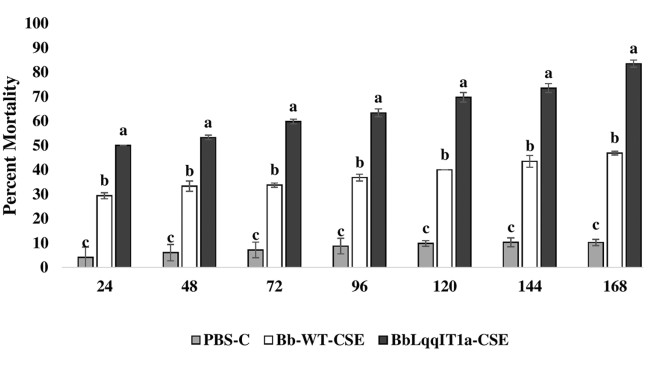

**Figure 6 Mortality of first instar nymphs of *A. craccivora* after treatment of crude soluble toxin extract (CSE) @ 5 µg/ml.** The error bars represents the standard error of the mean, different lower-case letters at the top of each bar shows significant differences using Duncan's multiple range test (DMRT) at 5% level

**Table 6** $LT_{50}$ of crude soluble toxic extracts of *B. bassiana* against *A. craccivora* first instar nymphs after 72 h *via* oral bioassay.

| Treatments @ 5 µg/ml of protein | $LT_{50}$ (h) | Slope ± SE | Fiducial (95%) (LL–UL) | $\chi^2$ value | *P*-value |
|---|---|---|---|---|---|
| PBS-C | 85.48 | 5.707 ± 0.72 | 13.2–179 | 3.094 | 0.213 |
| Bb-WT-CSE | 51.75 | 0.99 ± 0.182 | 19.98–134.015 | 5.94 | 0.0513 |
| BbLqqIT1a-CSE | 17.69 | 0.95 ± 0.16 | 7.25–43.17 | 7.93 | 0.019 |

Note:
Values are presented in Mean ± SE (standard error of the mean), df = 2, lower $LT_{50}$ equivalent to higher pathogenecity, $LT_{50}$, median lethal time (hours); fiducial limit at 95%; LL, lower limit; UL, upper limit; $\chi^2$, chi square value; $P \leq 0.05$ indicate significant differences among the treatment.

**Table 7** $LC_{50}$ of crude soluble toxic extracts of *B. bassiana* against *P. solenopsis* third instar nymphs after 168 h *via* oral bioassay.

| Treatments @ 5 µg/ml of protein | $LC_{50}$ (µg/ml) | Slope ± SE | Fiducial (95%) (LL–UL) | $\chi^2$ value | *P*-value |
|---|---|---|---|---|---|
| PBS-C | 79.7 | 3.88 ± 0.7 | 34.59–105.5 | 2.26 | 0.323 |
| Bb-WT-CSE | 36.77 | 0.708 ± 0.082 | 23.77–68.26 | 6.22 | 0.045 |
| BbLqqIT1a-CSE | 7.78 | 1.03 ± 0.082 | 6.20–9.96 | 9.31 | 0.009 |

Note:
Values are presented in Mean ± SE (standard error of the mean), df = 2, lower $LC_{50}$ equivalent to higher pathogenecity, $LC_{50}$, median lethal concentration; fiducial limit at 95%; LL, lower limit; UL, upper limit; $\chi^2$, chi square value; $P \leq 0.05$ indicate significant differences among the treatment.

**Table 8** $LC_{50}$ of crude soluble toxic extracts of *B. bassiana* against *A. craccivora* first instar nymphs after 168 h *via* oral bioassay.

| Treatments @ 5 µg/ml of protein | $LC_{50}$ (µg/ml) | Slope ± SE | Fiducial (95%) (LL–UL) | $\chi^2$ value | *P*-value |
|---|---|---|---|---|---|
| PBS-C | 85.48 | 5.707 ± 0.72 | 13.2–179 | 1.86 | 0.4 |
| Bb-WT-CSE | 9.87 | 0.68 ± 0.076 | 7.063–14.64 | 6.65 | 0.036 |
| BbLqqIT1a-CSE | 2.39 | 0.887 ± 0.81 | 1.76–3.104 | 6.39 | 0.04 |

Note:
Values are presented in Mean ± SE (SE= standard error of the mean), df = 2, lower $LC_{50}$ equivalents to higher pathogenecity, $LC_{50}$, median lethal concentration; fiducial limit at 95%; LL, lower limit; UL, upper limit; $\chi^2$, chi square value; $P \leq 0.05$ indicate significant differences among the treatment.

$P < 0.05$) (Table 6). When compared to the Bb-WT-CSE, the $LC_{50}$ values of BbLqqIT1a-CSE got reduced by about 4.72 and four folds at 5µg/ml of extract concentration for *P. solenopsis* and *A. craccivora*, respectively (Tables 7 and 8).

## Effect of LqqIT1 on the hemolymphic potassium and sodium ions in *P. solenopsis*

Treatment of third instar nymph of *P. solenopsis* with BbLqqIT1a-CSE led to the fluctuation in sodium and potassium concentrations compared to Bb-WT-CSE and PBS control (Table 9). In case of PBS control and Bb-WT-CSE, ionic concentration decreased gradually, while treatment of BbLqqIT1a-CSE led to increase in potassium ions in by 9.22 from 3.53 meq/ml ($F_{2,6} = 215.02$, $P = 0.000$) at 48 h of treatment and subsequently decreased by 5.07 from 9.22 meq/ml at 72 h ($F_{2,6} = 48.702$, $P = 0.001$). A marked decline in

**Table 9 Effect of crude soluble toxic extracts of *B. bassiana* @ 5 µg/ml protein on hemolymphic inorganic cations (K$^+$ and Na$^+$) in *P. solenopsis*.**

| Treatments | K$^+$ (meq/ml) | | | | Na$^+$ (meq/ml) | | | |
|---|---|---|---|---|---|---|---|---|
| | 24 h | 48 h | 72 h | 96 h | 24 h | 48 h | 72 h | 96 h |
| PBS(C) | 4.15 ± 0.11[a] | 3.29 ± 0.16[b] | 1.89 ± 0.32[b] | 2.55 ± 0.39[b] | 4.62 ± 0.22[a] | 3.27 ± 0.17[a] | 2.16 ± 0.17[b] | 2.5 ± 0.39[b] |
| Bb-WT-CSE | 4.43 ± 0.16[a] | 3.44 ± 0.09[b] | 1.5 ± 0.318[b] | 1.49 ± 0.22[c] | 4.07 ± 0.03[a] | 3.303 ± 0.24[a] | 1.18 ± 0.194[c] | 1.78 ± 0.34[b] |
| BbLqqIT1a-CSE | 3.57 ± 0.14[a] | 9.22 ± 0.33[a] | 5.07 ± 0.39[a] | 8.15 ± 0.34[a] | 1.25 ± 0.04[b] | 0.57 ± 0.31[b] | 3.92 ± 0.55[a] | 8.27 ± 0.35[a] |
| Df | 2.6 | 2.6 | 2.6 | 2.6 | 2.6 | 2.6 | 2.6 | 2.6 |
| CD @ 0.05 | 0.68 | 0.55 | 0.83 | 0.46 | 0.89 | 0.78 | 1.38 | 0.79 |
| F cal | 30.06 | 215.02 | 48.702 | 200.82 | 57.6 | 26.01 | 35.27 | 85.14 |
| *P*-value | 0.122 | 0.000 | 0.001 | 0.001 | 0.001 | 0.001 | 0.001 | 0.001 |

Note:
Mean ± SE (SE = standard error of the mean), every mean value represents the average of the three independent replications, means in vertical column with same superscript letters are not significant at 5% level of significance by Duncan's multiple range test (DMRT). Df, degree of freedom; CD, critical difference at 0.05; $P \leq 0.01$ indicate significant differences among the treatment.

sodium concentration was observed after 48 h of treatment. There was significant increase of sodium concentration as compared to the control after the 72 and 96 h of treatment.

### Effect of LqqIT1 on the hemolymphic potassium and sodium ions in *A. craccivora*

A significant drop was measured in the level of potassium ion (7.52 to 0.47 meq/ml) ($F_{2,6} = 127.8$, $P = 0.001$) due to the exposure of BbLqqIT1a-CSE at 96 h of treatment ($F_{2,6} = 5.68$, $P = 0.0004$). Likewise sodium ion level due to BbLqqIT1a-CSE gradually drop (3.05 to 0.62 meq/ml) ($F_{2,6} = 22.98$, $P = 0.002$) at all the time intervals (Table 10).

## DISCUSSION

### Improved pathogenicity of *B. bassiana* due to integration of LqqIT1 neurotoxin

The findings clearly demonstrated that the integration of LqqIT1 peptide in *B. bassiana* genome enhanced the infection ability against the *A. craccivora* by shortening the median lethal time. As an insect voltage gated sodium channel toxin, LqqIT1 causes paralysis and severe diseases in insects. At a concentration of $1 * 10^7$ spore/ml, EPF, *B. bassiana* stacked with the recombinant gene 'LqqIT1' from scorpion was found to be lethal to *A. craccivora* and mild toxic to *P. solenopsis*. The peptide sequence LqqIT1a is an excitatory insect specific toxin from the Buthidae family scorpion *Leiurus quinquestriatus quinquestriatus* (Ehrenberg), with 70 amino acid residues and no methionine or tryptophan (*Murugan & Saini, 2019*). By inducing repetitive firing in the voltage-gated sodium channels (VGSCs), this toxin causes flaccid paralysis in targeted insects (*Zlotkin et al., 1985*). The AaIT protein is an insect-specific neurotoxin from the buthid scorpion, *Androctonus australis* (Linnaeus) with 70 amino acid residues that has been used to improve the efficacy of EPF *M. anisopliae*. The LC$_{50}$ for *M. anisopliae* expressing AaIT was 22, 9, and 16-fold lesser in *Manduca sexta* (Linnaeus), *Aedes aegypti* (Linnaeus), and *Hypothenemus hampei* (Ferrari), respectively, than wild-type *M. anisopliae* infections in those insect species (*Wang & St Leger, 2007*; *Pava-Ripoll et al., 2008*). Similarly, *B. bassiana* expressing the AaIT

**Table 10 Effect of crude soluble toxic extracts of *B. bassiana* @ 5 µg/ml protein on hemolymphic inorganic cations (K$^+$ and Na$^+$) in *A. craccivora*.**

| Treatments | K$^+$ (meq/ml) | | | | Na$^+$ (meq/ml) | | | |
|---|---|---|---|---|---|---|---|---|
| | 24 h | 48 h | 72 h | 96 h | 24 h | 48 h | 72 h | 96 h |
| PBS(C) | 2.49 ± 0.24[c] | 1.2 ± 0.42[c] | 1.62 ± 0.2[b] | 0.57 ± 0.12[b] | 2.45 ± 0.296[c] | 1.2 ± 0.45[c] | 1.66 ± 0.39[c] | 0.6 ± 0.24[b] |
| Bb-WT-CSE | 4.14 ± 0.109[b] | 3.32 ± 0.2[a] | 2.79 ± 0.24[a] | 1.23 ± 0.4[a] | 4.37 ± 0.33[a] | 3.64 ± 0.25[a] | 4.35 ± 0.4[a] | 1.34 ± 0.34[b] |
| BbLqqIT1a-CSE | 7.52 ± 0.26[a] | 2.43 ± 0.33[b] | 0.69 ± 0.20[c] | 0.47 ± 0.23[b] | 3.05 ± 0.55[b] | 2.28 ± 0.38[b] | 2.66 ± 0.23[b] | 0.62 ± 0.205[a] |
| Df | 2.6 | 2.6 | 2.6 | 2.6 | 2.6 | 2.6 | 2.6 | 2.6 |
| CD @ 0.05 | 0.997 | 0.57 | 0.89 | 0.58 | 0.506 | 0.39 | 0.62 | 0.52 |
| F cal | 127.8 | 37.8 | 19.53 | 5.68 | 28.5 | 75.22 | 32.92 | 22.98 |
| *P*-value | 0.001 | 0.001 | 0.002 | 0.0004 | 0.001 | 0.002 | 0.001 | 0.002 |

**Note:**
Mean ± SE (SE = standard error of the mean), every mean value represents the average of the three independent replications, means in vertical column with same superscript letters are not significant at 5% level of significance by Duncan's multiple range test (DMRT). Df, degree of freedom; CD, critical difference at 0.05; $P \leq 0.01$ indicate significant differences among the treatment.

demonstrated 15-times improvement in the insecticidal activity in *Dendrolimus punctatus* (Walker) (*Lu et al., 2008*). According to our findings, the LT$_{50}$ of *B. bassiana* was reduced by 5.58 and 1.76 times, after leaf treatment and spray technique, respectively in *A. craccivora*. Similarly, the median lethal time of genetically modified *M. anisopliae* (MaLqqIT1-7) was reduced by 2.83 and 3.06 times against the *S. litura* and *A. craccivora*, respectively at 10$^7$ conidia/ml fungal concentration (*Kumar et al., 2023*). However, the LT$_{50}$ of *L. lecanii* expressing *BmKit* from *Buthus martensii* (Karsch) reduced by 26.5% against *Aphis gossypii* (Glover) (*Xie et al., 2015*). This probably may be due to the different mode of action of toxin molecules which acts preferably on different target sites in different insect pests.

It is well established that EPFs kill the host by breaching the cuticle and proliferates in the host's hemolymph (*St Leger & Wang, 2010*). Thus the pathogenicity of EPFs is primarily dependent on the proliferation of hyphal bodies (specialized yeast like cell phenotype) from the fungal mycelium in the host's hemocoel (*Pedrini, 2022*). In sap sucking insects, like, *A. craccivora*, microbial insecticides especially EPF work really effectively (*Purandare & Tenhumberg, 2012*). Laboratory trials demonstrated that transformed *B. bassiana* expressing 'LqqIT1' was not significantly virulent than the un-transformed parent strain against the *P. solenopsis*. The mean figures of cumulative mortalities of BbLqqIT1-3 and Bb-C were not significantly different (*P* = 0.122) in *P. solenopsis*. This suggests that the spores of BbLqqIT1-3 were unable to penetrate the waxy and hydrophobic cell wall of the *P. solenopsis*. On getting exposed to any foreign molecules (pathogen or protein molecule) host's body activates its immune system and mediates the detoxification mechanisms (*Serebrov et al., 2006*). Rapid ecdysis in *P. solenopsis* may be another important contributing factor for the poor pathogenicity of BbLqqIT1-3. However, the lower median lethal time (LT$_{50}$) of BbLqqIT1-3 indicated its boosted virulence against *A. craccivora*. As a result, LqqIT1 could be advocated as a promising candidate for improving the efficacy of other entomopathogens in general and EPFs in particular.

In cadavers, mycosis and sporulation appeared to be influenced by the mode of exposure and conidial concentration (*Tefera & Pringle, 2003*). The survival of spores on leaf surfaces until they encounter the host is ascribed to the leaf exposure approach, and the actual dose of conidia per insect is enhanced as the insect moves across the treated leaf surface (*Fernandez et al., 2001*). Compared to the spray method (46.67%), the leaf treatment method (53.37%) had the highest mean cumulative mortality at 96 h. In contrast, spray method caused higher mortality in *Aphis nerii* (Boyer de Fonscolombe) as compared to the leaf dip method (*Shivakumara, Keerthi & Polaiah, 2022*). This could be explained by the fact that moving on treated leaves while feeding is probably the most important means of contact leading to infection in case of contact pesticides.

## Entomotoxic activity of crude soluble toxic extract

Apart from their efficacy, several EPFs produced a variety of metabolically active toxin compounds that impede the defense mechanisms of the host (*Zhang & Xia, 2009*; *Gibson et al., 2014*). Insecticidal activity was previously reported for *M. anisopliae* and *B. bassiana* crude extracts and their purified entomotoxic fractions when given or injected to several insect pests (*Sahayaraj & Tomson, 2010*; *Freed et al., 2012*; *Keerio et al., 2020*, *Tomson et al., 2021*). The current study exhibited the sub-lethal effect of crude soluble toxic protein (CSE) extracted from transformed *B. bassiana* against *P. solenopsis* and *A. craccivora* and its effect on the concentration of hemolymphic Na$^+$ and K$^+$ levels in them. As we know, *P. solenopsis* and *A. craccivora* are both sap sucking insects, hence the crude soluble extract was applied to the leaf surface superficially. In the case of phloem feeders, oral exposure to insecticides can result in rapid mortality since the toxin molecules reach the target region in the insect's body immediately. In comparison to the topical spray method, the insect received the same dosage of toxicant in the dietary toxicity test (*Sadeghi et al., 2009*). Fungal derived crude extract are fully loaded with bioactive molecules, perhaps these bioactive chemicals helps to increase cellular permeability and leads to ionic leakage as the membranes are destroyed. EPFs have been reported to secret a wide range of bioactive metabolites in liquid culture or *in vitro*. *Beauveria* spp. are known to produce toxic metabolites including organic acids (oxalic acid), polyketides (oosporein), macrolactones (cephalosporolides), alkaloids (tennelin, bassianin, beauversetin, *etc.*), and cyclic depsipeptides (beauvericins, beauverolides, *etc.*), (*Strasser, Vey & Butt, 2000*; *Quesada-Moraga & Vey, 2004*). The composition of the liquid culture filtrate (crude toxins) from *B. bassiana* is complex. A cyclodepsipeptide (bassianolide) from *B. bassiana* and *V. lecanii* were reported as the main bioactive molecule (*Ortiz-Urquiza et al., 2010*). However, the main insecticidal substance in the crude soluble extract and the mechanism by which CSE was able to kill the *P. solenopsis* and *A. craccivora* require further investigations.

At 5μg/ml crude toxin concentration, the mean mortality rate of *P. solenopsis* for the BbLqqIT1-CSE was 80% at the conclusion of treatment, but the highest mortality was 83.34% in *A. craccivora* for the BbLqqIT1-CSE. Similarly, earlier findings have revealed the strong sub-lethal and toxic effects of crude toxins isolated from *M. anisopliae* and *B. bassiana* (un-transformed) on phloem feeders such as *Dysdercus cingulatus* (Fabricius) (*Sahayaraj & Tomson, 2010*), *A. gossypii* (*Gurulingappa, McGee & Sword, 2011*), *Plutella*

*xylostella* (L.) (*Freed et al., 2012*), *Bemisia tabaci* (Gennadius) (*Keerio et al., 2020*). Likewise, *Tomson et al. (2021)* reported 61.7% mortality in *P. solenopsis* by the application of fractionized fungal filtrate of *B. bassiana*. After 10 days of treatment, the transformed *B. bassiana* fungal strain (BbLqqIT1-3) could cause only 39% mortality in *P. solenopsis* at $1 * 10^9$ spore/ml; and the survived 61% nymphs could live for further 14 days (data not presented). At the end of treatment, *P. solenopsis* infected with the parent strain (un-transformed) Bb-C caused 46.7% mortality at 7 days after treatment, whereas crude extract of the same strain caused 53.33% mortality at 5μg/ml after 3 days of treatment. The BbLqqIT1-CSE was toxic with $LT_{50}$ values of 22.84 and 17.69 h for *P. solenopsis* and *A. craccivora*, respectively. As a result of the findings, it is clear that BbLqqIT1a-CSE was more lethal than recombinant BbLqqIT1-3 strain. If the conditions are unfavorable for spore germination, the crude soluble extracts may be an excellent alternative for biological control.

Inorganic constituents *viz.*, Na$^+$, K$^+$, and Ca$^{++}$ are essential because of its significance in the insect's neurophysiology and its levels inside and outside the nerve membrane, must be maintained for the impulse propagation (*Roeder, 1953*). It is well known that after being infected with pathogens, the level of these cations is being dramatically stimulated, and small changes in the level of hemolymphic inorganic ions can disrupt the cellular homeostatic environment and intracellular pH. Sudden change in the ionic concentration of hemolymph was responsible for its increased pH leading to general paralysis by *B. thuringiensis* in *Plodia interpunctella* (Hubner) (*Aboul-Ela et al., 1991*). A significant decline in sodium and potassium concentration was observed in *Spodoptera littoralis* (Boisduval) due to the treatment of *B. thuringiensis* (*Salama, Sharaby & Ragaei, 1983*). However, we observed that the levels of potassium and sodium ions in the third instar nymph of *P. solenopsis* got fluctuated at most of the observed time. At 48 h results showed significant increase of potassium ions and after 72 h of treatment it significantly decreased compared to control. But after 96 h, the concentration of both ions (Na$^+$ and K$^+$) was gradually and significantly increased in *P. solenopsis*. No significant changes observed in sodium and potassium ion concentration in *A. craccivora* treated with BbLqqIT1-3. As per our knowledge, no data is available on the effect of crude soluble toxin extract from transformed EPF and other insecticides on the levels of inorganic ions in the hemolymph of hemipteran insects, but other order such as lepidoptera is well studied (*Tiwari & Mehrotra, 1981*; *El-aziz, Nahla & Fahmy, 2015*).

## CONCLUSIONS

Present study documents about the boosted pathogenicity of recombinant strain (BbLqqIT1-3) against the *A. craccivora* and high insecticidal activity of crude soluble toxic extract (BbLqqIT1a-CSE) derived from engineered *B. bassiana* against the *P. solenopsis* and *A. craccivora*. Since only the enhanced efficacy of BbLqqIT1-3 was studied against the *P. solenopsis* and *A. craccivora* under in-vitro conditions, future studies are inevitable in order to ascertain the effectiveness and stability of engineered EPF/scorpion neurotoxin under the field/environmental conditions to prove that the genetically transformed *B. bassiana* (BbLqqIT1-3) is toxic to insect pests in general or to a selective group of insect

pests. Toxicity through oral mode is of direct practical implication in the natural condition. Furthermore, characterization and identification of these CSE would pave the way for developing effective formulations for sustainable insect pest management. For better understanding of a mode of action, histo-pathological studies must be performed in near future. A field evaluation of the toxic effects of the CSE from BbLqqIT1-3 would provide useful information for the bio-rational plant protection initiatives.

## ACKNOWLEDGEMENTS
The authors are highly thankful to Head, Division of Entomology Pusa Campus, ICAR-IARI, Pusa, New Delhi for providing facilities to conduct the trials.

### Funding
This work was financially supported by the University Grant Commission (CSIR-UGC) New Delhi, India in the form of a Junior Research Fellowship. The funders had no role in study design, data collection and analysis, decision to publish, or preparation of the manuscript.

### Grant Disclosures
The following grant information was disclosed by the authors:
 University Grant Commission (CSIR-UGC) New Delhi, India in the form of Junior Research Fellowship.

### Competing Interests
The authors declare that they have no competing interests.

### Author Contributions
- Sonam Rajput conceived and designed the experiments, analyzed the data, prepared figures and/or tables, authored or reviewed drafts of the article, and approved the final draft.
- Sachin S. Suroshe Suroshe conceived and designed the experiments, performed the experiments, analyzed the data, prepared figures and/or tables, authored or reviewed drafts of the article, and approved the final draft.
- Purnmasi Ram Yadav analyzed the data, prepared figures and/or tables, and approved the final draft.
- Anoop Kumar analyzed the data, prepared figures and/or tables, and approved the final draft.
- Gurvinder Kaur Saini analyzed the data, prepared figures and/or tables, and approved the final draft.

### Data Availability
 The raw data are available in the Supplemental File.

## Supplemental Information

Supplemental information for this article can be found online at http://dx.doi.org/10.7717/peerj.16030#supplemental-information.

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
