# Peer review of "Bioefficacy of engineered Beauveria bassiana with scorpion neurotoxin, LqqIT1 against cotton mealybug, Phenacoccus solenopsis and cowpea aphid, Aphis craccivora"

_PeerJ, doi:10.7717/peerj.16030_

## Round 0.1 · original submission · Major Revisions

The three reviewers found your manuscript suitable for consideration for publication in PeerJ. Before publication, however, several questions raised by reviewers need to be addressed. Consequently, I recommend major revisions for your manuscript.

Reviewer 1 ·

Basic reporting

The implication of this study could be the potential development of a novel approach for controlling cowpea aphids (Aphis craccivora) using genetically transformed Beauveria bassiana fungus expressing a recombinant toxin gene from a scorpion. The study suggests that the transformed fungus (BbLqqIT1-3) was highly pathogenic against cowpea aphids, causing significant mortality and displaying neurotoxicity symptoms in the treated insects.

Experimental design

1. Studies performed in the last five years are missing in this manuscript.
2. Please include a detailed objective of this study and with what hypothesis “the transformation of bioagent with neurotoxin was performed”.
3. It would be helpful to include information on the economic impact or ecological significance of these pests

Validity of the findings

4. The text briefly mentions the cloning of a toxin gene from a scorpion and its expression in the entomopathogenic fungus Beauveria bassiana. However, it does not provide details about the specific genetic modification techniques used or the process of transformation. More information on the methodology would enhance the clarity and reproducibility of the study.
5. The manuscript lacks details on the specific properties and composition of the toxin extracts.

·

Basic reporting

The manuscript is written using clear and unambiguous, professional English language easily understandable by the readers.
The authors have provided enough explanations on the background of the experimental needs, provided extensive literature review synthesized in a meaningful way.
The article structure is adherent to the standard format advocated by the journal.

Few minor copyediting/typographical corrections are needed before final acceptance of MS for publication.

Experimental design

The experimental designs and statistical analyses applied to the data generated are appropriate and adequate one.
The adequate and standard procedures have been adopted for performing experiments.
The manuscript falls well in the scope of journal.
The existing knowledge gap of efficacy improvement of entomopathogenic fungi has been well addressed by the present study.

Validity of the findings

Authors have meaningfully interpretated and discussed their results. All the data generated in the present study have been provided in the tables and or figures as per the requirement.

The authors draw valid conclusions based on the significant findings of their research work.

Additional comments

The authors of MS titled "Bioefficacy of engineered Beauveria bassiana with scorpion neurotoxin, LqqIT1 against cotton mealybug, Phenacoccus solenopsis and cowpea aphid, Aphis craccivora" have done remarkable work on pathogenicity enhancement of EPF Beuveria bassiana engineered with scorpion toxin.

The increased virulence and reduced lethal times are the promising achievements for further development of promising bioformulations towards biointensive management of agriculturally important crop pests.

The manuscript has a scientific merit in terms of its valid background, sound methods and meaningful findings having potential applicability in devising ecofriendly pest management options. Therefore, the MS deserves publication in PeerJ, provided few queries mentioned below are addressed by the authors in revisions.

1. What is the reason for high specifity of transformed/ engineered EPF agaisnt A. craccivora over P. solenopsis? I could not find any explanations in discussion section. The detailed information highlighting this aspect in discussion section of revised version will benefit the readers.

2. How will you ensure the stability of engineered EPF/ scorpion neurotoxin under field/ environmental condition, which may have serious effect of the efficacy against target pest insect?

Reviewer 3 ·

Basic reporting

Bioefficacy of engineered Beauveria bassiana with scorpion neurotoxin, LqqIT1 against cotton mealybug, Phenacoccus solenopsis and cowpea aphid, Aphis craccivora describes the effect of bioengineered entomopathognenic fungus for the management of sucking insect pests.
The manuscript needs improvement of (English)
References are sufficient but are inconsistent.

Experimental design

The manuscript has described well the experimental design, but the question is why DMRT was used?

Validity of the findings

The findings are novel in a sense that the biocontrol agents are being genetically modified to control the insect pests. It will not reduce the application of insecticides pressure, but will also safe the environment and life from the threats of insecticides bio-magnification etc.

Annotated reviews are not available for download in order to protect the identity of reviewers who chose to remain anonymous.

---

## Round 0.2 · accepted · Accept

The authors have revised the manuscript according to the reviewer's comments. Now It can be accepted in its current form.

Reviewer 1 ·

Basic reporting

The manuscript is improved based on suggested lines.

Experimental design

Design is appropriate.

Validity of the findings

Findings are correct and well explained.

Additional comments

In future publications, authors are suggested to give detailed response in the rebuttal files rather than stating "added".

Reviewer 3 ·

Basic reporting

No Comment

Experimental design

The experimental design is well defined

Validity of the findings

The findings are valid for the use of microorganisms for the control of insect pests

Additional comments

There are minor mistakes which need correction. Plz see the annotated file.

Annotated reviews are not available for download in order to protect the identity of reviewers who chose to remain anonymous.